# Can We Improve the Functional Threshold Power Test by Adding High-Intensity Priming Arm-Crank?

**DOI:** 10.3390/jfmk6040088

**Published:** 2021-10-28

**Authors:** Dmitri Valiulin, Priit Purge, Peter Hofmann, Jarek Mäestu, Jaak Jürimäe

**Affiliations:** 1Institute of Sport Sciences and Physiotherapy, Faculty of Medicine, University of Tartu, 51008 Tartu, Estonia; priit.purge@ut.ee (P.P.); jarek.maestu@ut.ee (J.M.); jaak.jurimae@ut.ee (J.J.); 2Training & Training Therapy Research Group, Institute of Human Movement Science, Sports & Health, Exercise Physiology, University of Graz, 8010 Graz, Austria; peter.hofmann@uni-graz.at

**Keywords:** warm-up exercise, lactic acid, anaerobic threshold, athletic performance

## Abstract

The aim of the present study was to evaluate the effects of arm-crank induced priming on subsequent 20 min Functional Threshold Power Test among 11 well-trained male cyclists (18.8 ± 0.9 years; 182 ± 5 cm; 73.0 ± 6.6 kg; *V˙*O_2max_ 67.9 ± 5.1 mL·kg^−1^·min^−1^). Participants completed an incremental test and two maximal performance tests (MPTs) in a randomized order. Warm-up prior to MPT_low_ consisted of 20 min aerobic exercise and 25 s high-intensity all-out arm crank effort was added to warm-up in MPT_high_. Constant intensities for the first 17 min of MPT were targeting to achieve a similar relative fatigue according to participants’ physiological capacity before the last 3 min all-out spurt. Final 3 min all-out spurt power was 4.94 ± 0.27 W·kg^−1^ and 4.85 ± 0.39 W·kg^−1^ in MPT_low_ and MPT_high_, respectively (not statistically different: *p* = 0.116; *d* = 0.5). Blood lactate [La] levels just before the start were higher (*p* < 0.001; *d* = 2.6) in MPT_high_ (5.6 ± 0.5 mmol·L^−1^) compared to MPT_low_ (1.1 ± 0.1 mmol·L^−1^). According to *V**˙*CO_2_ and net [La] data, significantly higher anaerobic energy production was detected among MPT_low_ condition. In conclusion, priming significantly reduced anaerobic energy contribution but did neither improve nor decrease group mean performance although effects were variable. We suggest priming to have beneficial effects based on previous studies; however, the effects are individual and additional studies are needed to distinguish such detailed effects in single athletes.

## 1. Introduction

An adequate warm-up is a key aspect of successful and sustainable performance with better perception and lower risk of traumas [1]. Finding a balance between warm-up induced activation and physiological benefits remains crucial for every pre-competitive preparation. In this context, maintaining the muscles in an optimal pre-start condition plays an important role before each competition [1]. Although higher warm-up intensities can produce a greater priming effect, the selected intensity should not induce fatigue among athletes. Therefore, for a beneficial outcome from the warm-up, it is suggested to apply an individual approach, as the recovery time after warm-up may vary depending on the individual [2,3].

In addition to increased body and muscle temperature, other systemic changes such as metabolic, neural, and psychological conditioning are increasingly recognised as beneficial to optimise performance [1]. According to previous studies, physiological systemic benefits, such as accelerated oxygen uptake kinetics (*V˙*O_2_) and inhibited lactate production [4], can be induced by distant muscles, categorised as non-primary for a particular sport [5,6]. Such priming is carried out on a systemic level and can be achieved despite smaller working muscle mass or short effort duration. Studies have shown that pre-loading by the arms can be compared to prior leg sprint exercise [6,7] and induce similar *V˙*O_2_ levels up to 45 min after high-intensity exercise [8]. Despite clear evidence for physiological changes, there have also been contradictory findings stating that non-target muscle priming does not have a beneficial effect on subsequent performance [9,10]. Possible reasons pointed to were inhibited anaerobic energy contribution, overly intensive warm-up, and insufficient recovery after warm-up, as well as modified pacing strategies [9,10]. In contrast, performance-enhancing effects of priming have been prescribed for small muscle group exercise [5] and researchers are still seeking to define the optimal protocol [9,10].

High-intensity performance requires a high maximal oxygen uptake (*V˙*O_2max_) for endurance athletes, but the true challenge is to sustain a high fraction of *V˙*O_2max_ for a prolonged time [11]. *V˙*O_2max_ alone was not shown to be a good predictor of endurance performance when athletes of similar endurance ability are compared [12]. In fact economy at first (VT1) and second (VT2) ventilatory thresholds play a crucial role, since athletes with nearly identical *V˙*O_2max_ values can perform at different levels [11].

Aerobic training often improves submaximal thresholds such as VT1 and the VT2 as well as other physiological variables without a concomitant increase in *V˙*O_2max_ [13]. The VT2 representative of the maximal lactate steady state (mLSS), has a superior relationship to endurance performance compared to *V˙*O_2max_ and is suggested to be a better indicator of aerobic endurance [13]. VT2 and mLSS describe a maximal metabolic steady state in which lactate production and utilisation are in a fine systemic balance [14] and improvements of VT2 were shown to enhance performance [15].

Studies have shown that athletes are capable of maintaining VT2 intensity for an impressive amount of time. Some research limited the capacity to 60 min [16,17] but others raised discussions about the accuracy of this statement, questioning its physiological basis [18]. Considering athletes’ individuality and the different research methods, work duration at mLSS intensity is suggested to be about ~20–60 min but may be dependent on exercise mode and muscle mass involved [16,17]. However, as the power increases above VT2 intensity, the duration dramatically decreases. Identifying the critical point of maximal metabolic steady state and predicting performance capability is a main concern for many athletes, coaches, and sports scientists. Therefore, VT2 and *V˙*O_2max_ are both used to select a tailored exercise workload [19,20].

Functional threshold power (FTP) is a frequently used additionally important performance quality indicator [21] which is highly correlated to mLSS [21] and relative power exertion during a mass-start bike race [22]. It is the maximum mean power, which can be sustained for an approximately 1 h period and is a good predictor of overall performance capacity [21]. Denham and colleagues [21] found that FTP can be evaluated by a time efficient 20 min FTP test (FTP20), which predominantly relies on aerobic metabolism [21]. Considering previous suggestions to test upper-body high-intensity pre-loading on more aerobic distances [10], we decided to use FTP20 to be as cycling-specific as possible for our evaluation method.

The aim of current study was to develop an applicable warm-up protocol, that athletes would benefit from during the competition. Secondly, to investigate the effect of upper-body priming on subsequent primarily aerobic performance. Open-end final spurt was included as a competitive component, considering suggestions about longer distance, lower overall intensity, and longer recovery phase before performance (>10 min) [10]. We hypothesised that MPT_high_ participants have increased performance and *V˙*O_2_ kinetics.

## 2. Materials and Methods

### 2.1. Participants

Eleven competitive well-trained male cyclists (mean ± SD 18.8 ± 0.9 years; 182 ± 5 cm; 73.0 ± 6.6 kg; *V˙*O_2max_ 67.9 ± 5.1 mL·kg^−1^·min^−1^) (Table 1) volunteered to participate in this study. Inclusion criteria were regular cycling at competitive level, male gender, and age between 18 and 30 years. Exclusion criteria were chronic health issues, disease, or trauma. The procedures of this study were approved by the local university ethics committee (290/T-17) in accordance with the Declaration of Helsinki. All participants were required to give their written informed consent at the first laboratory visit. Participants were instructed to arrive at the laboratory in a rested and adequately hydrated state at least 2 h postprandial. They were asked to maintain their normal diet and avoid ingesting alcohol and caffeine 24 h before each laboratory visit. Moreover, only moderate training loads the day before each test and maximal efforts at least 3 days before the laboratory visit were allowed.

### 2.2. Experimental Overview

Participants attended the laboratory on three occasions within a 3-week period, with each visit being separated by at least 3 recovery days. On the first visit participants’ body composition was measured using dual-energy X-ray absorptiometry (DEXA) (Hologic Discovery DXA; Massachusetts, USA) and they were introduced to the testing equipment. To exclude any health risks associated with the maximum stress, all participants had to perform a specialist supervised incremental cycle ergometer test as the first exertion. In addition, they were asked to try out manual switching of power intensity with ± 5 W and ± 50 W after completion of the incremental cycle ergometer test, to accustom them to the Cyclus2 ergometer (RBM Elektronik-Automation GmbH; Leipzig, Germany). On two subsequent occasions maximal performance tests (MPT) were performed. Gas exchange variables, heart rate and blood [La] concentration were measured in all tests.

### 2.3. Incremental Test

On the first visit participants completed a maximal incremental cycle ergometer exercise test starting at 100 W and increments of 30 W every third minute. Incremental and experimental tests were performed on a cycle ergometer Cyclus-2. The apparatus was recently calibrated, and participants used their own bikes which was supported by the system. The workload was increased until participants were no longer able to sustain the load. Heart rate was measured continuously in all tests and was stored and analysed in 1 s intervals with a Polar H7 sensor (Polar Electro Oy; Espoo, Finland). Spirometric breath-by-breath measures (Cortex Metamax 3B, Cortex Biophysik; Leipzig, Germany) were measured and analysed. Laboratory ambient air temperature was 22 °C and relative humidity 40%, all measurements were done at the same time of the day for each participant. Two gas exchange thresholds (VT1, VT2) were determined by an experienced researcher considering a disproportionate increase in *V˙*CO_2_ relative to *V˙*O_2_, ventilation breakpoints (V-slope), visual inspection of individual plots and respiratory-exchange-ratio (RER) value. The highest average measured O_2_ during the 30 s period was considered *V˙*O_2max_. The intensity that would require 10% Δ [VT2 plus 10% difference between the work rate at VT2 and work rate achieved at the end of the incremental test (WR_peak_)] was subsequently calculated [19,20].

### 2.4. Experimental Tests

In order to objectively compare performance in two maximal cycling tests under upper-body pre-load or common warm-up conditions, participants performed both protocols in a randomised order. Participants were instructed to complete both maximal tests as quick as possible.

MPT_low_ and MPT_high_ (Figure 1) protocols started with a similar 20 min warm-up at 40% of P_max_. A 25 s high-intensity all-out arm crank effort (braking weight: 35 g·kg^−1^ body weight) on an upper-body hand crank ergometer (Monark Ergomedic 849E; Vansbro, Sweden) was added for the MPT_high_ protocol. In both testing conditions, participants were required to recover from the warm-up for a minimum of 10 min plus self-determined duration, until they reported readiness for the subsequent maximal performance test. The 20 min maximal performance bouts started with 17 min of constant work at a controlled pace of 10% Δ VT2 to cover the possible standard error from determining VT2 and ensure a gradual increase in blood [La] concentration [18]. During the final 3 min of the FTP20 performance, athletes were required to perform their self-paced maximal final spurt. They were provided with a 3 s countdown before the start and instructed to increase the pace to achieve the maximal exertion by the end of the 20 min.

Strong verbal encouragement was provided during both maximal tests. Remaining time was the only information that cyclists were provided, and participants were made unaware of implemented power (W); the power display was covered to exclude any numerical comparison.

Blood [La] concentration (EKF-Diagnostic; Barleben, Germany) was determined for both conditions at rest (baseline), after the warm-up (immediately after, + 3′, + 5′), after the anaerobic priming load (immediately after, + 3′, + 4′, + 5′, + 7′), during the test (immediately before, + 5′, + 10′, + 15′), and during recovery (immediately after MPT, + 3′, + 4′, + 5′, + 7′, + 9′, + 11′, + 13′, + 15′) (total of 21 samples in MPT_high_ and 16 in MPT_low_ protocol). Arterialised blood samples (20 μL) were taken from a pre-warmed fingertip. The finger was always cleansed with alcohol, and the first drop of blood was removed to prevent contamination of the sample. At the same time as the [La] measurements, participants were asked to evaluate their overall and muscle fatigue using the 20-point Borg scale [23].

### 2.5. Statistical Analyses

Data management and analysis were performed using IBM SPSS Statistics Data Editor 23.0 (New York, USA). The distribution of each variable was examined for the assumption of normality using the Shapiro–Wilk test, visual inspection of descriptive statistics, and z-score. All measured data was considered normally distributed. Parametric methods such as paired samples *t*-test and 2-way repeated measures ANOVA test with *p* < 0.05 as the level of significance were used to explore, investigate the data, and compare groups, because all measured data was normally distributed. Correlations were determined using Pearson’s correlation coefficient (r). The magnitudes of the correlation coefficients were stratified into groups comprising negligible (*r* < 0.30), low (0.30 < *r* < 0.50), moderate (0.50 < *r* < 0.70), high (*r* > 0.7) [24]. Data are expressed as mean ± standard deviation (SD), presented with effect size (η^2^) and confidence intervals (CI). Effect size of Cohen *d* = 0.2 was considered as ‘small’, *d* = 0.5 ‘medium’, and *d* = 0.8 ‘large’ [25]. Partial Eta squared for two variables in repeated measures η^2^ < 0.04 was considered ‘small’, η^2^ = 0.25 ‘moderate’, and η^2^ > 0.64 ‘strong’ [26].

A prior power analysis was performed using G*Power© software (version 3.1.9.2, 2017) for comparison between two independent means. This was based on previously reported effect sizes 1.6 [9] and 1.7 [27], an alpha criterion of 0.05, and power of 0.8. Beforehand analysis indicated that a total of 7 participants were required to reach 0.8 statistical power with an aim of 4.7 and 5.0 W·kg^−1^ with standard deviation ± 0.2.

## 3. Results

### 3.1. Self-Paced Finish

Participants completed both tests with initial 17 min mean power of 4.2 ± 0.5 W·kg^−1^. Last 3 min self-paced finish MPT_low_ mean power was 4.94 ± 0.27 and 4.85 ± 0.39 W·kg^−1^ in MPT_high_, respectively, which was not statistically different (*p* = 0.116; *d* = 0.5; CI_95%_ = [−0.0, 0.3]) (Figure 2). Strong correlations occurred between MPT_low_ self-paced finish power and incremental test values of *V˙*O_2max_ (mL·kg^−1^·min^−1^) (*p* = 0.005; *r* = 0.78), maximal relative power (W·kg^−1^) (*p* = 0.011; *r* = 0.73), and 10% Δ VT2 (*p* = 0.016; *r* = 0.70). Self-paced finish power was not correlated with self-selected recovery time after warm-up, even though recovery time after warm-up was significantly longer in the MPT_high_ condition as the strenuous priming exercise significantly delayed the feeling of readiness from 813 ± 138 s in MPT_low_ to 958 ± 226 s in MPT_high_ (*p* = 0.011; *d* = 0.9; CI_95%_ = [42, 249]). A detailed data set regarding splits is presented in Table 2.

### 3.2. Blood Lactate Concentration

The MPT_high_ group was subjected to 25 s all-out arm crank exercise (mean power 259 ± 42 W), which elevated [La] to 8.1 ± 0.3 mmol·L^−1^ in the sixth minute after arm-crank exercise. Although participants recovered until full readiness to perform, [La] levels before the start were still elevated at 5.7 ± 0.5 mmol·L^−1^ and significantly higher in MPT_high_ compared to MPT_low_ at 1.1 ± 0.1 mmol·L^−1^ (*p* < 0.001; *d* = 2.6; CI_95%_ = [3.4, 5.7]) (Figure 3 and Figure 4). Priming arm-crank exercise and longer recovery time significantly influenced the subsequent time course of [La], as [La] values changed differently over time (*p* < 0.001; η^2^ = 0.51). Calculating the anaerobic contribution for each performance showed a significantly higher net [La] increase per mean W·kg^−1^ in MPT_low_ during the first 5 min (*p* < 0.001; *d* = 1.6), the second 5 min (*p* = 0.008; *d* = 1.0), and the final 5 min of performance (*p* = 0.019; *d* = 0.8). Moreover, the interaction between calculated anaerobic values was also significant (*p* = 0.003; η^2^ = 0.37), meaning body metabolic rates differed over time between both performances. Despite the differences in metabolic rates, no correlations were found between the last 5 min [La] and mean power during the final spurt (*p* > 0.05; η^2^ < 0.50).

### 3.3. Spirometry

Although an all-out arm crank priming exercise was imposed in MPT_high_, the self-selected recovery was sufficient to erase the elevated *V˙*O_2_ level. The values at the start did not differ between MPT_high_ with 0.59 ± 0.12 L·min^−1^ compared to 0.56 ± 0.13 L·min^−1^ in MPT_low_ (*p* = 0.640; *d* = 0.1; CI_95%_=[−0.11, 0.17]). Between-group effects were also insignificant (*p* = 0.57; η^2^ = 0.73) throughout the tests. Calculation of metabolic economy was conducted by relating *V˙*O_2_ to mean power for each split but showed no significant difference between groups throughout the tests (*p* > 0.05).

*V˙*CO_2_ values were not different at the start between MPT_high_ with 0.53 ± 0.11 L·min^−1^ compared to 0.53 ± 0.13 L·min^−1^ in MPT_low_ (*p* = 0.96; *d* = 0.02) and the between-group effect was insignificant (*p* = 0.15; η^2^ = 0.15) throughout the test. RER values demonstrated a steady-state (Figure 4) from minutes 10 to 17 of the maximal performance up to the final spurt and reached its peak by the end of performance with no significant between-group effect (*p* = 0.39; η^2^ = 0.10). A visualisation of spirometric values shown in Figure 5.

### 3.4. Perceived Exertion

Reported muscle type readiness which felt convenient to start the exertion was indicated by 8.1 ± 1.7 points in MPT_high_ compared to 7.1 ± 1.4 in MPT_low_ (*p* = 0.031) according to the Borg 20-point scale. Reported overall readiness was found insignificantly (*p* = 0.126) higher at 8.0 ± 1.5 points in MPT_high_ compared to 7.1 ± 1.4 points in MPT_low_, respectively. Between-group effects during 15 min of recovery after the maximal performance were insignificant for overall fatigue (*p* = 0.80; η^2^ = 0.02) and muscle fatigue (*p* = 0.12; η^2^ = 0.01), meaning no subjectively determined benefit related to recovery could be pointed to in either case.

## 4. Discussion

The aim of this study was to investigate the effect of upper-body priming exercise prior to a 20 min Functional Threshold Power Test. Returning to the hypothesis posed at the beginning of this study, it is now possible to state that starting the race with elevated [La] from the same fatigue level as the non-primed group neither showed a beneficial nor an impairing effect regarding relative power or recovery although net lactate increase was clearly diminished in MPT_high_, also no difference in *V˙*O_2_ kinetics could be observed.

The findings of the current study are consistent with those of Baker and colleagues [28] who found that aerobic metabolism and energy production are able to support extremely high muscle force application and power outputs—meaning that a preceding high-intensity bout or a preliminary race will not necessarily impair a subsequent time trial.

As the race progresses, aerobic energy production becomes more dominant. Figure 3 presents an overview of [La] changes throughout the MPTs. [La] change during the first 5 min of MPT_high_ was significantly lower, which means that the same performance could be delivered with less anaerobic metabolism. If an increased demand for aerobic energy production could be less strenuous than anaerobic production, then lower overall fatigue during the start phase could be maintained. Apparently, this gives some space for higher intensity at the start, although this is not supported by a recent study [10]. Moreover, from Figure 3 it is obvious that in MPT_low_ a markedly higher anaerobic energy contribution during the final 5′ of the race could be performed, although [La] levels after 15 min were not significantly different between both interventions (*p* = 0.53; *d* = 0.2). We suggest that participants are used to a gradual increase of [La] such as in MPT_low_, whereas in MPT_high,_ [La] values already plateaued at significant levels and anaerobic energy contribution may have been already depleted/inhibited to some part by the beginning of the final spurt. Although FTP20 is predominantly aerobic [21], possibly the duration was still not long enough to bring out beneficial effects of priming.

The current findings make it clear that a poor pacing strategy [9] cannot be the reason for priming a non-beneficial effect as both groups were paced to the same fatigue levels using the same strategy and a fixed pace. During the last 3 min, a cycling competition reaches its culmination where no pacing can be rationalised as an all-out strategy is most common. Pacing should instead be timed at greatest physiological response achieved by priming and should be used as a tool to realise the potential. The 5 min split analysis did not indicate significant difference between spirometric variable levels, although additional analyses showed statistical differences in *V˙*O_2_, RER, and VE parameters during the first 2 min of MPT between both groups. In combination with a suitable choice of intensity a significant improvement may be suggested due to the priming.

Correlation analysis showed that MPT_low_ was correlated with variables from the incremental tests such as *V˙*O_2max_ (mL·kg^−1^·min^−1^), maximal relative power (W·kg^−1^), and 10% Δ VT2 for the first 17 min. Since MPT_high_ did not correlate with any of these values, it is suggested that performance becomes less predictable after priming exercise and can easily diminish previous training. However, many studies [7,19,20,29] have shown the physiological advantages of priming, although they cannot guarantee any improved results during an actual race [30]. In this case, we still suggest pre-load to have beneficial effects, which are highly individual and rather empirical methods are needed to distinguish such effects in every single athlete. For example, four participants (26.7%) benefited from the priming and developed higher mean power during last 3 min of the final spurt in our study. A possible explanation for this might be the neuromuscular excitability, which was induced on central level, since priming was performed with non-primary muscle groups. However, previous studies prescribed neuromuscular potentiation mainly to single repetition or sprint exercise, but endurance sport on submaximal level is not common [3].

Furthermore, the specific sport should be re-evaluated as maybe the priming produced by bigger muscle mass can elevate [La] levels to a greater extent without causing significant fatigue. Reverse circumstances compared to our study such as applied by Birnbaumer et al. [5] may be more favourable to successfully increase performance.

In our study, aerobic energy contribution was facilitated at the expense of anaerobic energy. A more economical power production due to a primed aerobic metabolism could be beneficial in case of an energy deficit during longer distances among comparable athletes and saving more energetic reserves for the competition state. Perceived exertion data, overall fatigue, and peak heart rate lower than HR_max_ indicated that the FTP20 is too short to induce energy depletion. Furthermore, Borg values showed statistically significant changes for reported leg muscle fatigue levels at determined readiness to perform, but at the same time reported overall fatigue did not show any statistical difference. Certainly, the practical significance of a 1-point difference is not highly relevant, although it can give an idea of altered perception after priming and that athletes are less dependent on muscle fatigue rather than overall fatigue when determining readiness to perform. This can give some confidence for further studies to feel free to use higher loads for priming effects as muscle alterations cannot be the primary reason to limit readiness to perform.

The findings of this study are subject to at least two limitations. Although power calculation showed our sample size to be enough, total number of 15–20 participants would allow us to use regression analysis, which could be even more informative. Secondly, some of the participants did not switch power during the final all-out spurt although preferring to add more watts.

## 5. Conclusions

In summary, priming effect is considered smaller than previously suggested and no substantial performance enhancing warm-up protocol could be developed from the current study design. Although priming does decrease anaerobic energy production which in turn promotes aerobic energy production, these changes neither improved nor decreased competitive performance.

## Figures and Tables

**Figure 1 jfmk-06-00088-f001:**
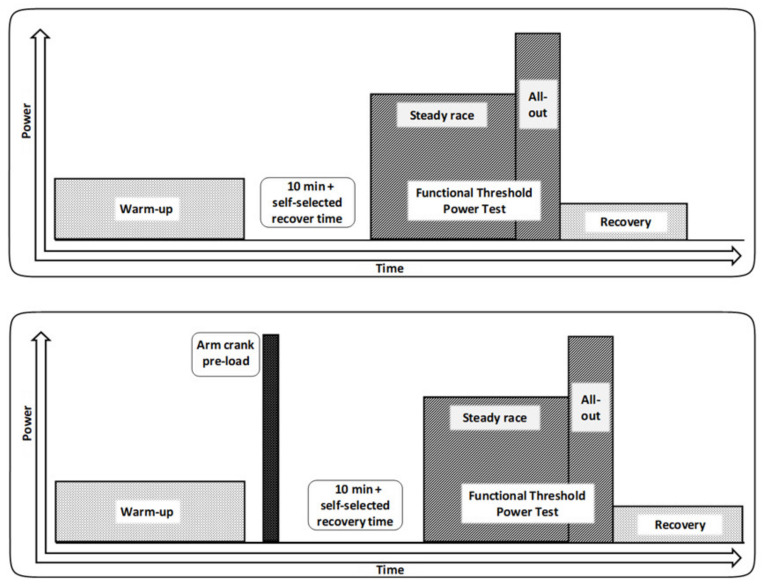
Maximal performance test-low protocol (MPT_low_) and high protocol (MPT_high_).

**Figure 2 jfmk-06-00088-f002:**
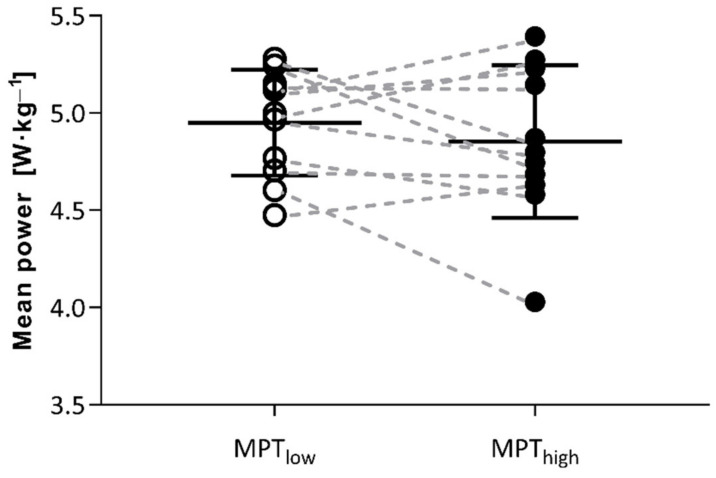
MPT performance (power output) in final spurt of MPT_low_ and MPT_high_.

**Figure 3 jfmk-06-00088-f003:**
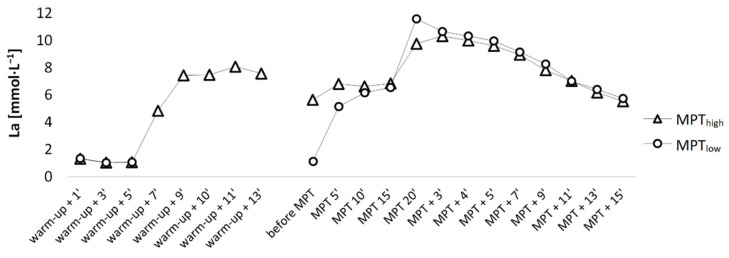
[La] response to both protocols.

**Figure 4 jfmk-06-00088-f004:**
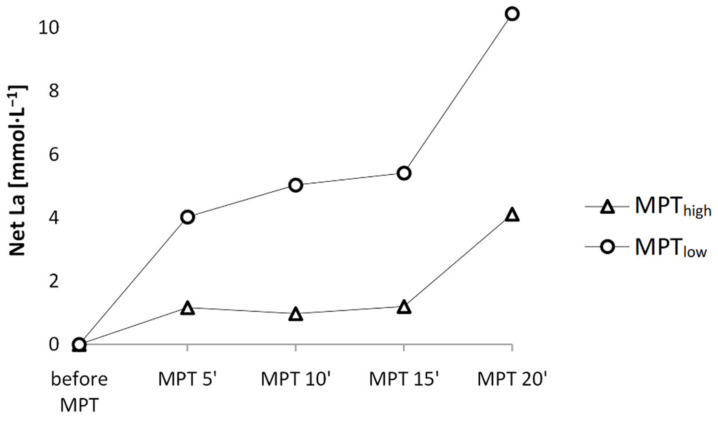
Net [La] increase during MPTs.

**Figure 5 jfmk-06-00088-f005:**
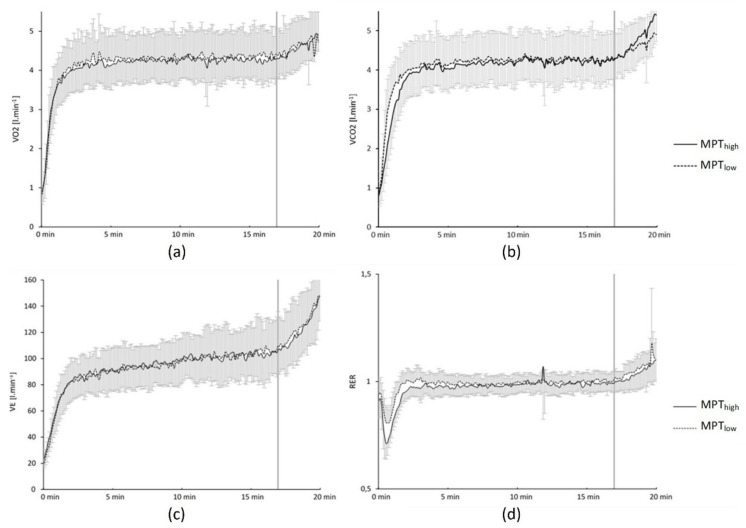
(**a**) Oxygen uptake (*V˙*O_2_); (**b**) carbon dioxide output (*V˙*CO_2_); (**c**) ventilation (*V˙*E); (**d**) respiratory exchange ratio (RER), after a usual low intensity warm up (MPT_low_) and an additional 25 s all-out arm crank priming pre-load (MPT_high_).

**Table 1 jfmk-06-00088-t001:** Participants’ characteristics (*n* = 11).

Characteristic	Mean ± SD
Age (years)	18.8 ± 0.9
Height (m)	1.8 ± 0.0
Weight (kg)	73.0 ± 6.6
Total lean mass (kg)	57.5 ± 4.9
Leg lean mass (kg)	20.8 ± 2.1
BMI (kg·m^−2^)	22.0 ± 1.9
*V˙*O_2max_ (mL·kg^−1^·min^−1^)	67.9 ± 5.1
*V˙*O_2max_ (L·min^−1^)	5.0 ± 0.6
P_max_ (W)	369.0 ± 47.9
Ventilatory threshold VT1 (W)	181.4 ± 43.3
Ventilatory threshold VT2 (W)	295.6 ± 39.9

BMI—body mass index; *V˙*O_2max_—maximal oxygen consumption.

**Table 2 jfmk-06-00088-t002:** Physiological changes throughout MPTs (*n* = 11).

Stage	Parameter	MPT_low_	MPT_high_	*p*-Value	Effect Size (95% CI)
Warm-up	Mean power (W·kg^−1^)	147.6 ± 19.1	147.6 ± 19.1		
	Recovery time (s)	813 ± 138	958 ± 226	0.011 *	0.94 (41.7 to 249.2)
Before start	*V˙*O_2_ (L·min^−1^)	0.6 ± 0.1	0.6 ± 0.1	0.640	0.14 (−0.1 to 0.2)
	*V˙*CO_2_ (L·min^−1^)	0.5 ± 0.1	0.5 ± 0.1	0.955	0.02 (−0.1 to 0.1)
	[La] (mmol·L^−1^)	1.1 ± 0.4	5.6 ± 1.7	< 0.001 **	2.67 (3.4 to 5.7)
MPT 5′	*V˙*O_2_ (L·min^−1^)	3.7 ± 0.6	3.7 ± 0.5	0.434	0.25 (−0.1 to 0.3)
	*V˙*CO_2_ (L·min^−1^)	3.9 ± 0.8	3.4 ± 0.5	0.098	0.55 (−0.1 to 1.1)
	[La] (mmol·L^−1^)	5.2 ± 0.8	6.8 ± 1.1	< 0.001 **	1.55 (0.9 to 2.4)
	Net [La] increase (mmol·L^−1^)	4.0 ± 0.8	1.2 ± 0.7	< 0.001 **	1.72 (1.7 to 4.0)
MPT 10′	*V˙*O_2_ (L·min^−1^)	4.3 ± 0.6	4.2 ± 0.5	0.580	0.17 (−0.1 to 0.2)
	*V˙*CO_2_ (L·min^−1^)	4.2 ± 0.6	4.2 ± 0.5	0.413	0.26 (−0.1 to 0.3)
	[La] (mmol·L^−1^)	6.2 ± 1.1	6.6 ± 1.8	0.191	0.42 (−0.3 to 1.2)
	Net [La] increase (mmol·L^−1^)	1.2 ± 0.2	−0.2 ± 1.6	0.011 *	0.93 (0.4 to 2.4)
MPT 15′	*V˙*O_2_ (L·min^−1^)	4.3 ± 0.7	4.3 ± 0.5	0.451	0.23 (−0.1 to 0.3)
	*V˙*CO_2_ (L·min^−1^)	4.3 ± 0.7	4.2 ± 0.5	0.331	0.31 (−0.1 to 0.3)
	[La] (mmol·L^−1^)	6.6 ± 1.5	6.9 ± 2.1	0.530	0.20 (−0.7 to 1.3)
	Net [La] increase (mmol·L^−1^)	0.4 ± 0.8	0.2 ± 0.9	0.713	0.11 (−0.8 to 1.1)
MPT 20′	*V˙*O_2_ (L·min^−1^)	4.6 ± 0.6	4.6 ± 0.5	0.804	0.08 (−0.2 to 0.3)
	*V˙*CO_2_ (L·min^−1^)	4.9 ± 0.6	4.8 ± 0.6	0.456	0.23 (−0.2 to 0.4)
	[La] (mmol·L^−1^)	11.6 ± 2.7	9.8 ± 3.4	0.073	0.60 (−0.2 to 3.8)
	Net [La] increase (mmol·L^−1^)	5.0 ± 2.1	2.9 ± 2.3	0.015 *	0.88 (0.5 to 3.7)
Performance	Average power_steady_ (W·kg^−1^)	4.2 ± 0.5	4.2 ± 0.5		
	Average power_all-out_ (W·kg^−1^)	4.94 ± 0.27	4.85 ± 0.39	0.308	0.32 (0.1 to 0.3)
	[La_net_] (mmol·L^−1^)	10.8 ± 2.9	5.4 ± 2.6	< 0.001 **	2.16 (−7.1 to−3.7)
	[La_max_] (mmol·L^−1^)	11.9 ± 2.8	11.1 ± 2.7	0.200	0.41 (−0.5 to 2.3)

MPT_low_—maximal performance test without prior loading; MPT_high_—maximal performance test with prior loading; CI—confidence interval of the difference; *—*p* < 0.05; **—*p* ≤ 0.001.

## Data Availability

Data are contained and available within this manuscript.

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
