# Peer review of "Can We Improve the Functional Threshold Power Test by Adding High-Intensity Priming Arm-Crank?"

_jfmk, 2021, doi:10.3390/jfmk6040088_

Round 1

Reviewer 1 Report

The paper is reasonably well written, but some improvements are required throughout. There should be more discussion about the potential mechanisms of priming exercise benefiting subsequent exercise performance. I would think that the neuromuscular aspects should be featured with changes in muscle activation. Need to be consistent with the abbreviations and units reported. As an example VO2max is ml.kg.min. Other comments are listed below.

Line 18: Spell out blood lactate before abbreviating.

Line 39: “…to optimise performance.”

Lines 43-45: Please revise this sentence. Not quite sure what you are trying to state?

Line 56: “…play a crucial role,”

Line 61: “…compared to VO2max..”

Line 61: “…to be a better indicator of..”

Line 70: “…muscle mass involved.”

Line 83: “…would benefit from during…”

Line 87: MPT? But you have described FPT?

Line 92: ml.kg.min

Table 1: Total lean mass.

Table 1: ml.kg.min

Line 108: Spell out DEXA before abbreviating it.

Line 126: “..end of the incremental test..”

Line 127: “Heart rate (HR)..”

Line 146: You now use the term “FTP”. Be consistent.

Figure 1: Not sure what constitutes the FTP?

Line 154: Replace “landmark” with “information”

Line 157: More information about the precise timings for blood lactate would be helpful.

Lines 166-168: Was data normally distributed?

Line 174: The sign used for effect size is that of partial eta.

Line 177: Please check the moderate and large partial eta because they seem rather large.

Line 296: This part of the sentence is difficult to follow “…after priming exercise easily diminishing previous training.” Please revise.

Author Response

We thank the reviewers for their constructive and valuable reports and revised our manuscript accordingly. All detailed changed can be followed in the manuscript (track changes) and in the supplemental table. We addressed all issues point-by-point and kindly request to continue the review process.

Reviewer 2 Report

Dear Authors,

I was pleased to review the manuscript entitled "Can we improve the Functional Threshold Power test by adding high-intensity priming arm-crank?" and to commend the authors for their research.
The study aim was to find an applicable warm-up protocol, that athletes would benefit during the competition. 
The study is well designed and structured. The manuscript is well written so I only have minor concerns to comment and suggest:
# line 20 please correct the abbreviation of VCO2 max and it is suggested to put square brackets to express the blood lactate concentration [La]
# line 44, 53, 53 and rest of manuscript - please change the abbreviation of VO2 and VO2max
Methodology
# Since there were respiratory gas measurements, please indicate ambient conditions of temperature and relative humidity
# Please also indicate the period of the day when measurements were done
# please clarify the reasons for the drop in the number of subjects from fifteen to eleven
# It is suggested that the graphics of figure 2 be made uniform with those of figures 3, 4 and 5 
After the authors make the suggested changes, I am of the opinion that the manuscript is suitable for publication, respecting the standards and quality criteria of the J. Funct. Morphol. Kinesiol.

BW

JB

Author Response

(The authors gave the same response as above.)

Reviewer 3 Report

Thank you for the opportunity to review the manuscript. 

Abstract:

Line 10: The aim is not clear

Line 12: Was the incremental test maximal?

Line 13: Is the o maximal performance tests the same as FTP test?

Line 19: It is difficult to understand the methods. What about incremental test?

Line 23: The conclusion was not supported by the results.

Introduction

Line 29: needs reference

Line 43: as you write studies, you need more than 1 reference.

Line 55: reference

Line 56: Review this reference. The concepts of running economy and thresholds are mixed.

Line 62: caution should be taken with this affirmation. Review the reference.

Line 75: as you are talking about MLSS, it is important to write about the correlation between FTP and MLSS.

Line 75: it is important to include the concept of FTP.

Line 82: “to find a protocol” How did you look for?

Line 83: It is not clear

Line 87: What is MPT?

Materials and Methods

Line 90: replace subjects to participants

Line 100: Include inclusion and exclusion criteria

Table 1: Do you have 15 (in the text) or 11 (in the table) participants?

Line 110: Describe the text.

Line 121: Were gas exchange thresholds not checked by another person?

Line 125: without plateau?

Line 126: It is not clear

Line 129: Start the section describing the equipment used.

Line 131: is the equipment reliable?

Line 169: It is important to make clear for which comparisons each test was used.

Line 169: Post hoc?

Table 2: Some data are presented in Table and in figure. It is not necessary.

Line 247: When the Perceived Exertion was measured?

Discussion section can be enhanced after the introduction and methods correction.

Discuss the study hypothesis.

Include the study limitations.

The conclusions are not answering the initial aim.

The aims written at the end of the introduction are not the same as in the discussion.

Author Response

(The authors gave the same response as above.)

Round 2

Reviewer 1 Report

Well done on addressing my comments.

Reviewer 3 Report

Thank you for your corrections.